# Purine Nucleotides Metabolism and Signaling in Huntington’s Disease: Search for a Target for Novel Therapies

**DOI:** 10.3390/ijms22126545

**Published:** 2021-06-18

**Authors:** Marta Tomczyk, Talita Glaser, Ewa M. Slominska, Henning Ulrich, Ryszard T. Smolenski

**Affiliations:** 1Department of Biochemistry, Medical University of Gdansk, 80-210 Gdansk, Poland; eslom@gumed.edu.pl; 2Department of Biochemistry, Institute of Chemistry, University of São Paulo, São Paulo 05508-000, Brazil; talita.glaser@usp.br (T.G.); henning@iq.usp.br (H.U.)

**Keywords:** purine metabolism, purinergic signaling, Huntington’s disease

## Abstract

Huntington’s disease (HD) is a multi-system disorder that is caused by expanded CAG repeats within the exon-1 of the huntingtin (*HTT*) gene that translate to the polyglutamine stretch in the HTT protein. HTT interacts with the proteins involved in gene transcription, endocytosis, and metabolism. HTT may also directly or indirectly affect purine metabolism and signaling. We aimed to review existing data and discuss the modulation of the purinergic system as a new therapeutic target in HD. Impaired intracellular nucleotide metabolism in the HD affected system (CNS, skeletal muscle and heart) may lead to extracellular accumulation of purine metabolites, its unusual catabolism, and modulation of purinergic signaling. The mechanisms of observed changes might be different in affected systems. Based on collected findings, compounds leading to purine and ATP pool reconstruction as well as purinergic receptor activity modulators, i.e., P2X7 receptor antagonists, may be applied for HD treatment.

## 1. Introduction

### 1.1. Huntington’s Disease Pathophysiology

Huntington’s disease (HD) is a rare neurodegenerative disease that extensively affects the central nervous system. The disorder is inherited in an autosomal dominant manner. Clinically, HD is manifested by the occurrence of cognitive, mental, and motor disorders [1]. One of the earliest signs of the motor disorder in Huntington’s disease is chorea, i.e., involuntary dance-like movements. Patients with HD are also characterized by bradykinesia (motor slowness) and dystonia (the occurrence of unnaturally slow, prolonged muscle spasms that cause repetitive torsional movements affecting various parts of the body) [2]. Thus, the motor disorders in Huntington’s disease visibly affect the attitude, balance, and gait of HD patients. Furthermore, with the development of the disease, the speech of a patient with HD becomes unclear. Moreover, swallowing difficulties may also occur, which may lead to weight loss [3]. In addition to movement disorders, HD causes prominent changes in personality and mood. Most often HD patients suffer from depression, apathy, anxiety, irritability, outbursts of anger, impulsiveness, obsessive-compulsive syndromes, sleep disorders, and withdrawal from social life [4]. A characteristic feature of Huntington’s disease is also cognitive impairment, which affects the understanding, reasoning, and memory. It includes slower thinking, problems with concentration, organization, planning, decision-making, answering questions, short-term memory disorders, as well as limited problem-solving skills and understanding of new information [2]. The incidence of HD in Europe is estimated at 5 to 10 cases per 100,000 people. In adults, the first symptoms appear between 30 and 50 years of age, after which the disease relentlessly progresses over the next 15–20 years.

The genetic cause of HD is the occurrence of multiple repeats of the CAG nucleotide sequence within the huntingtin gene (*HTT*) localized on chromosome 4, which results in the elongation of the polyglutamine stretch in the HTT protein. The number of CAG nucleotide sequence repeats in the healthy population varies from 6 to 35, while the presence of over 36 repeats defines the pathogenic HD allele. In cases with 36 to 39 CAG repeats within the *HTT* gene, the symptoms of the disease may be reduced or completely unnoticeable [1]. Moreover, the expansion length of CAG repeats correlates with the onset of the disease [5]. Huntingtin (HTT) is a multi-domain protein with a size of 348 kDa, with the highest level of HTT, demonstrated in the brain [6,7]. It has also been demonstrated outside the nervous system, in organs such as skeletal muscles or the heart [3]. The elongation of the polyglutamine stretch in exon 1 HTT leads to the formation of insoluble huntingtin aggregates, which are observed in both the early and advanced stages of the disease [8]. Aggregates of the mutated form of HTT (mHTT) have been identified in the brain as well as the outside central nervous system, e.g., in skeletal muscle [9]. In the CNS, mHTT mainly affects the basal ganglia region of the encephalon; this is the main region for voluntary and involuntary motor control, as well as cognition. This mutant protein sensitizes GABAergic neurons, making them vulnerable to NDMA induced excitotoxicity, leading to cell death. On the cellular level, HTT was found in the nucleus, endoplasmic reticulum, Golgi apparatus, and endosomes [10,11,12]. It has been shown that HTT interacts with proteins involved in gene transcription (e.g., CREB-binding transcription factor (CBP)), intracellular signaling (e.g., HIP14 protein), intracellular transport (e.g., HIP1 protein, HAP1), endocytosis, and metabolism (e.g., PACSIN1 phosphoprotein, vitamin D-binding receptor, hepatic X-receptor) [13,14]. Furthermore, HTT is essential during early embryogenesis and brain development. The inactivation of the *HTT* gene by targeting exon 1 or 5 is lethal in mice on embryonic day 7.5 (E7.5) of mouse development [15]. Biochemical and molecular pathways by which mutant huntingtin affects cellular dysfunction and death remain unclear; however, these might be caused not only by cellular mHTT accumulation but also the loss of HTT function leading to metabolic and signaling cascades impairment. Thus, in this work, we aimed to summarize the knowledge about the dysfunction of intra- and extracellular metabolism related to purines in the most affected by Huntington’s disease systems (central nervous system, heart, skeletal muscle), its role in HD pathophysiology, and possible applications in HD treatment.

### 1.2. Purine Nucleotides Metabolism and Signaling

Purines play an important role as metabolic signals, controlling cellular growth and providing energy to the cell. In the central nervous system (CNS), the balance of nucleotides depends on a continuous supply of preformed purine and pyrimidine rings, mainly in the form of nucleosides. These nucleosides can enter the brain through the blood–brain barrier, or locally supplied by the conversion of extracellular phosphorylated forms (nucleotides) by extracellular nucleotidases located in the neuronal plasma membrane. The ectonucleotidases are divided into four families that differ in the specificity of the substrate and cellular location: nucleoside triphosphate diphosphohydrolases (NTPDases), nucleotide pyrophosphatase/ phosphodiesterases (NPPs), alkaline and acid phosphatases (ALP and ACP, respectively), and ecto-5′-nucleotidase [16,17,18,19]. The NTPDase comprises NTPDase1–8; however, just NTPDase1, -2, -3, and -8 can efficiently hydrolyze all nucleotides. The NPP family includes seven members (NPP1–7) but as NTPDASE, only NPP1, NPP2, and NPP3 can hydrolyze nucleotides [17]. The ALP and ACP families comprise many ectoenzymes that dephosphorylate nucleotides (ATP, ADP, and AMP) and diverse substrates. The human 5′-nucleotides family has seven enzymes, although just one is anchored to the plasma membrane, known as CD73 [19,20]. Its main function is the production of extracellular adenosine. Later in the extracellular cascade, this adenosine can be converted to inosine through ecto-adenosine deaminase (eADA), and later to hypoxanthine by purine nucleoside phosphorylase (PNP) [21]. Then, after the transport of nucleosides and inosine/hypoxanthine into the cell, they are converted to AMP, ADP, and ATP by the basic cellular processes similar to those taking place in muscles.

In skeletal muscles and the heart, high energy phosphate produced in oxidative phosphorylation is transported from mitochondria to the contractile apparatus via phosphocreatine (PCr) shuttle. In the mitochondrial inter-membrane space, the energy of the high-energy phosphate bond of ATP can be transferred to creatine by mitochondrial creatine kinase (CK) resulting in the formation of PCr. In the cytosol, PCr can be used to resynthesize ATP from ADP by cytosolic CK. An important aspect of ATP involvement in energy metabolism is ATP degradation to adenosine-5′-diphosphate (ADP) by ATPases (e.g., CK, sodium–potassium, or calcium myosin ATPase). There is also a possibility of further conversion of ADP to AMP that is mediated by adenylate kinase (AK). AMP is a substrate for two alternative pathways and enzymes: (1) 5′-nucleotidase (5NT) dephosphorylating AMP to adenosine that occurs in multiple isoforms, and (2) AMP deaminase (AMPD) converting AMP to inosine monophosphate (IMP). A unique aspect of purine nucleotide metabolism in the skeletal muscle is the function of the purine nucleotide cycle that besides AMPD, involves also adenylosuccinate synthetase, and adenylosuccinate lyase. This cycle plays an important role in energy balance through the maintenance of a high ATP/ADP ratio. Higher levels of intracellular AMP may also activate the AMP-activated protein kinase, an important protein involved in the regulation of cellular energy metabolism at both protein expression and activity levels. IMP is also the final product of purine de novo synthesis as well as purine salvage pathway (formation of IMP from hypoxanthine). The adenosine can be degraded to inosine by adenosine deaminase (ADA). Afterward, inosine can be converted to hypoxanthine by a purine nucleoside phosphorylase (PNP). Hypoxanthine can be converted by xanthine oxidoreductase activity to xanthine and uric acid. Nucleotide breakdown is highly organ and cell-type specific.

In the extracellular space, ATP can act as a signaling molecule by interacting with purinergic P2X and P2Y receptors. While ADP, UTP, UDP, and UDP-glucose interact with P2Y receptor subtypes, P2X receptors are ligand-gated cation channels comprised of seven subtypes (P2X1-7) that assemble in a trimeric structure and upon stimulation allow cations inflow (such as Na^+^, K^+^, and Ca^2+^). P2Y receptors consist of eight subtypes (P2Y1, 2, 4, 6, 11, 12, 13, 14), in which P2Y1,2,4,6 and 11 receptors couple to Gq proteins, activating phospholipase C (PLC)-β. Resulting in IP3 and diacylglycerol production, releasing Ca^2+^ from intracellular stores [22]. Instead, P2Y12,13, and 14 receptors couple to Gi proteins, inhibiting cAMP production [22]. While adenosine is a ligand for a category of receptors named P1. This category is composed of A1, A2A, A2B, and A3 subtypes. Within, A1 and A3 receptors are coupled to Gi/Go proteins, inhibit adenylate cyclase and reduce cAMP levels upon activation; A2 receptors are coupled to Gs protein activating adenylate cyclase, increasing intracellular cAMP levels [23].

## 2. Purines in Huntington’s Disease

### 2.1. Cellular Changes Related to Affected Huntingtin Expression

In 2014, Ismailoglu et al. investigated the metabolic profile of three syngeneic mouse embryonic stem cell (mESC) lines: HTT knock-out (KO), extended poly-Q (Htt-Q140/7), and wildtype mESCs (Htt-Q7/7). They found that HTT KO cells exhibited a 50% decrease in ATP levels, concomitant with 2-fold increases in both ADP and AMP levels, which demonstrated that HTT protein activity is critical for the maintenance of high energy phosphates in the cell. Moreover, HTT KO exhibited unique expression of several molecules involved in purine synthesis, such as 5-aminoimidazole ribonucleotide (AIR), phosphor-ribosyl-formyl-glycineamidine (FGAM), formamidoimidazole-4-carboxamide ribotide (FAICAR), and 5′-phosphoribosyl-4-(N-succinocarboxamide)-5-aminoimidazole (SAICAR). It was accompanied by increased levels of the 5-aminoimidazole-4-carboxamide ribotide (AICAR) and the final pathway product of purine biosynthesis, IMP in those cells. Overall, it reveals a substantial acceleration of purine synthesis and turnover in HTT KO mESCs and suggesting the HTT importance in maintaining its mutual balance [24]. Following these findings, in 2020, we differentiated HTT KO mESCs to neurons as well as to cardiomyocytes and established that besides HTT absence, differentiation was successful [25,26]. It suggested that HTT is not a fundamental protein in cardiomyocyte development. The previous study in human pluripotent stem cells suggested that loss of HTT can reduce the induction of neural and neuronal genes during differentiation [27]. In the case of investigation purine metabolism, there were no changes in extracellular ATP concentration between HTT KO and WT neurons [25]. While HTT KO cardiomyocytes exhibited diminished intracellular ATP pool, which is in the line with data obtained from the mHTT overexpression cellular model [26,28]. HEK 293T cell line transfected with plasmids expressing the mutant exon 1 of the *HTT* gene was characterized also by increased ADA activity, which suggested deteriorations in intracellular purine metabolism. Increased intracellular levels of metabolites such as inosine, hypoxanthine, and adenosine were found in HTT KO mESC [24]. Besides changes in intracellular purine metabolism, HD cells exhibited reduced activities of all extracellular enzymes, including eNTPD, e5NT, and eADA relative to the control [28]. In the case of purinergic signaling, the Ca^2+^ response of P2 receptors is impaired in mouse HTT KO neurons derived from mESC [25], while the P2Y2 receptor response is only impaired in the presence of mHTT. Lack of P2Y2 receptor input induces the cells to upregulate their expression, as a feedback compensation effort [25].

### 2.2. Purine Nucleotides Metabolism and Signaling in the Central Nervous System in Huntington’s Disease

One of the pathological hallmarks of the HD-affected brain is the gradual atrophy of the striatum (caudate nucleus and putamen) [29]. On gross examination, 80% of HD brains show atrophy of the frontal lobes. Thus, bilateral, symmetric atrophy of the striatum is observed in 95% of the HD brains [29]. The mean brain weight in HD patients is approximately 30% lower than in normal individuals. Striatal degeneration may lead to energy metabolism changes. It is clear that neurons are highly dependent on mitochondria ATP and Ca^2+^ buffering to maintain synaptic communication [30]. Moreover, neuronal mitochondria levels require to renew or adapt by efficient biogenesis and mitophagy during their lifespan [31]. There are undisputed data that highlighted the intensive deficits in energy metabolism in the human HD-affected brain. The striatum mitochondrial oxidative metabolism investigation underlined the selective defect of glycolysis in early and clinical symptoms in HD patients [31].

Interestingly, further analysis showed a significant correlation between impaired basal ganglia metabolism and functional capacity of HD patients [32]. Studies concentrated on lactate metabolism in HD-affected brains are ambiguous. Increased lactate levels were observed in the striatum of HD patients, which is discussed as inefficient oxidative phosphorylation leading to lactate accumulation from pyruvate via lactate dehydrogenase [32]. In contrast, reduced levels of lactate and citrate were shown in cerebrospinal fluid which may indicate impairment of glycolysis and TCA cycle function in HD subjects [32].

At the molecular level, brain energy metabolism deterioration included mitochondria dysfunction and trafficking interruption resulted in changes in the activities of molecules involved in energy balance [33]. In few independent studies of the striatum of mHTT knock-in mice, HD patients’ postmortem brains, and lymphoblasts, the ATP/ADP ratio was reduced as a consequence of mHTT aggregation [34,35,36]. Significant reduction in mitochondrial spare respiratory capacity was reported in human HD fibroblasts and immortalized mHTT expressing mouse striatal cells when compared to wild-type cells indicating that mitochondrial bioenergetics is compromised by mHTT and supporting a toxic role of mHTT on mitochondrial bioenergetics [37].

Moreover, expression of full-length mHTT in immortalized striatal progenitor cells, derived from HD mice model *Hdh*Q111, diminished the activity of mitochondrial respiratory chain complex II associated with intense sensitivity to Ca^2+^-induced decrease in oxygen consumption and mitochondrial membrane potential [33]. Mitochondrial Ca^2+^ transport is powered by the mitochondrial proton gradient, and increased neuronal Ca^2+^ modifies mitochondrial ATP production by uncoupling oxidative phosphorylation. In healthy conditions, Ca^2+^ can promote ATP synthesis by assisting pyruvate dehydrogenase, isocitrate dehydrogenase, α-ketoglutarate dehydrogenase, and ATP synthase complex [31]. The reduced mitochondrial ATP levels and decreased ATP/ADP ratio found in mHTT-containing striatal cells is linked to increased Ca^2+^ influx through N-methyl-D-aspartate (NMDA) receptors, and cell ATP/ADP ratio is normalized by blocking Ca^2+^ influx [36]. Mitochondria incubated with mHTT had increased sensitivity to Ca^2+^-induced opening of the MPT and release of cytochrome c and apoptosis induction [38], consistent with a direct effect of mHTT on mitochondrial Ca^2+^ handling [33]. As demonstrated by mitochondria isolated from lymphoblast cells of HD patients and HD mouse brain that have a reduced membrane potential and depolarize at lower Ca^2+^ concentrations than control mitochondria [39]. Both wild HTT and mHTT bind to the outer mitochondrial membrane in human neuroblastoma cells and cultured striatal cells [40]. It is established that full-length mHTT may impair mitochondrial motility in neurons through a toxic gain of loss of function from the polyglutamine tract [40]. Moreover, in vitro and in vivo models of HD are characterized by the altered mitochondrial trafficking that precedes neuronal dysfunction [41].

Roles of mitochondria in HD go far beyond ATP production and Ca^2+^ homeostasis; they can also regulate the metabolism of the reactive oxygen species (ROS) and apoptosis [42]. In addition, mHTT can affect the production of neurotrophins, such as brain-derived neurotrophic factor (BDNF), impairing neuronal survival [43]. Noteworthy, BDNF promotes ATP synthesis and mitochondrial efficiency [44,45], which correlates with the fact that neurons are high-demand energy cells [46].

The reduction of mitochondrial bioenergetics in HD could be also a result of impairment of mitochondrial enzymes. Postmortem studies of the striatum of HD patients as well as cultured striatal neurons transfected with N-terminus mHTT showed selective depletion of succinate dehydrogenase associated with decreased complex II enzymatic activity [47,48]. The mHTT can interfere with the TATA box binding protein (TBP)-associated factor 4 (TAF4)/ cyclic adenosine monophosphate (cAMP) response element-binding protein (CREB) signaling pathway [39]. Data are underlining a reduction in CREB phosphorylation and CRE signaling, which may contribute to the down-regulation of molecules involved in energy metabolism, such as peroxisome proliferator-activated receptor-gamma coactivator 1 alpha (PGC-1α) in mHTT expressing striatal cells [34]. Moreover, the reduction in PGC-1α correlates with a diminished number of mitochondria in HD postmortem brain tissue [49,50]. The mHTT can impair the mitochondria biogenesis through PGC-1α transcription inhibition as well. Post-mortem HD brain analysis shows lower levels of PGC-1α [38,51]. In this way, cells increase the anaerobic metabolism in the basal ganglia of the HD patients, leading to improved lactate generation and its accumulation, thus promoting local inflammation [32].

AMP-activated protein kinase (AMPK), the main sensor for cellular energy content is also targeted by mHTT. It is activated by increased AMP/ATP ratios and induces PGC-1α expression. AMPK is located in the nucleus in the HD striatum, downregulating the Bcl-2 family which leads to apoptosis [52,53].

As previously mentioned, brain capacity for *de novo* production of purines and pyrimidine rings is limited. In this regard, the brain demands constant provision of nucleosides produced in the liver and that cross the blood–brain barrier [30]. Adenosine and ATP in the extracellular milieu of the encephalon stimulate P1 and P2 receptors, respectively, acting as co-neurotransmitters. ATP and UTP can be released to the extracellular environment by microglia, astrocytes, and neurons themselves. They are also present in vesicles, where concentrations can be as high as 100 mM (ATP) and 8 mM (UTP) [30]. Thus, the imbalance of intra/extracellular nucleotides in neurons is related to various neurological disorders, such as Parkinson’s, Alzheimer’s, and Huntington’s disease [54]. Figure 1 summarizes deteriorations in purine nucleotides metabolism and signaling in CNS affected by Huntington’s disease.

HD is characterized by GABAergic loss due to excessive calcium response to glutamate stimuli. However, some evidence points to the impairment of some metabolic pathways link with purines metabolism [55]. Patassini and coworkers, using human post-mortem brains, underwent metabolomics analyses of many brain areas typically affected in patients of Huntington’s disease [56]. They found a decreased level of hypoxanthine but an increased level of inosine, indicating higher activity of ADA and low PNP. On the other hand, another study with a transgenic mouse model (R6/1) of HD highlighted an increased adenosine level in the cerebrospinal fluid in the middle stage of HD [57,58]. Furthermore, microarray analysis of the prefrontal cortex of HD patients showed a significant reduction in the transcript of CD73, suggesting that the synaptic adenosine level converted from AMP might be low. Moreover, the transcript levels of ENTs and AKs in HD patients are higher than in non-HD subjects [59]. Thus, it seems that regulation of the adenosine modulating enzymes in HD is still unclear and requires further investigation.

So far, few studies have investigated the role of purinergic signaling in Huntington’s disease. Regarding the adenosine receptors, A1, which is expressed in neurons, microglia, astrocytes, and oligodendrocytes is impaired in Huntington’s disease, this receptor usually protects against degeneration by inhibiting excitatory neurotransmission. A2A receptor is the most studied in HD; however, this is still controversial. Some studies detected downregulated A2A receptor expression in HD rodent models [60,61,62,63,64], while others improved the motor symptoms by antagonizing the A2A receptor [65,66]. Regarding ATP-sensitive receptors, the P2X7 receptor has been described as a major player, since its antagonism by Brilliant Blue G or selective P2X7 receptor inhibitors mitigates dyskinesia and body weight loss while preventing neuronal loss in the Tet/HD94 and R6/1 models [67].

Besides the P2X7 receptor, ATP and UTP-sensitive P2Y2 receptor plays important roles in HD. The intracellular signaling triggered by this receptor is impaired in neural precursor cells and neurons of HD human and mouse in vitro models. Moreover, the activity of the P2Y2 receptor favors the differentiation of neural stem cells towards a GABAergic neuronal fate [25]. Reestablishment of the activity of the P2Y2 receptor, promoting BDNF release, may prevent cell death.

### 2.3. Purine Nucleotides Metabolism and Signaling in Skeletal Muscle and Heart in Huntington’s Disease

It has been shown that HD patients, except for the central nervous system disorders, are also characterized by a reduced (by about 50%) muscular strength compared to healthy subjects [68]. Moreover, HD mice models were characterized by skeletal muscle atrophy [69]. It is also noted that R6/2 mice have altered the ultrastructure of transverse tubules in skeletal muscle fibers [70]. At the cellular level, aggregation of mHTT, the inclusion of poly-ubiquitinated proteins were found in myofibers and myonuclei in R6/2 mice [9,71]. The mHTT formation in skeletal muscle leads to defects, such as myofiber size reduction, type switching, and denervation [69,72,73,74,75]. In addition to changes in myofiber structure, transcriptional deregulation, and deteriorations in energy metabolism occur linked to impaired adenine nucleotide metabolism [76]. It has been noted that the skeletal muscles of HD patients are characterized by dysfunction of oxidative metabolism [77]. Studies in experimental mouse models have also shown that mitochondria isolated from the quadriceps muscle of the R6/2 mice model were characterized by reduced activity of the respiratory chain complexes [78]. Increased production of energy substrates such as lactate and acetate were also shown which confirms the presence of oxidative metabolism disorders [79]. Furthermore, in vitro myocyte cultures revealed disturbances of the mitochondrial membrane potential and cytochrome c release [80]. Moreover, increased levels of the mitochondrial pro-fission factor DRP1 and its phosphorylated active form, and decreased levels of the pro-fusion factor MFN2 in quadriceps of the R6/2 mice model were detected [78]. Interestingly, experimental studies have also shown a reduction in the level of PGC-1α, one of the proteins activated by peroxisomal proliferator gamma (PPAR γ) in skeletal muscles of HD mice models as well as HD patients [50]. Additionally, the pharmacological activation of this co-activator led to increased expression of the skeletal muscle fiber proteins that suggested an important role of energy metabolism abnormalities in development of HD related myopathy [81]. Recently, Miller et al. report that the decrease of adenine nucleotides strictly linked with energy metabolism of the cell may lead to increased nucleotide degradation and contribute to the general pathophysiology of skeletal muscle atrophy in numerous disease states and conditions [82].

Besides skeletal muscle pathology, multiple epidemiological studies have shown that heart failure is the second cause of death in HD patients [83,84]. Reduced cortical and subcortical blood flow and heart rate have been reported as examples of pathological abnormalities in HD hearts [85,86,87,88]. HD patients’ heart rate variability pattern is consistent with a higher sympathetic prevalence [89]. Further studies with HD animal models reaffirmed cardiac pathological events, such as variations in the heart rate and cardiac remodeling [90,91,92]. HD mouse models also revealed heart contractile dysfunctions, which might be a part of dilated cardiomyopathy. These changes were accompanied by an increased expression of fetal genes (the same is observed during the pathological remodeling of the heart) or the presence of interstitial fibrosis [90]. Interestingly, hearts of the HD mouse model R6/2 did not react to the same extent upon long-term treatment with isoproterenol (a compound that causes hypertrophy of the heart), as wild-type mouse hearts, suggesting the presence of signaling dysfunction that stimulates heart remodeling [93]. Hearts of HD mice models contained an increased number of apoptotic cells and degree of fibrosis [94]. Cardiomyocytes of the BACHD HD mouse model also showed electromechanical abnormalities, including prolonged action potential or arrhythmic contractions. Cellular arrhythmia was accompanied by increased activity of Ca^2+^/calmodulin-dependent protein kinase II, suggesting disturbed calcium metabolism in the cell [95]. Dridi et al. show that intracellular calcium (Ca^2+^) leaks via post-translationally modified ryanodine receptor/intracellular calcium release (RyR) channels may play an important role in HD pathology [96]. Furthermore, abnormalities of superoxide dismutase activity and glutathione peroxidase in the mitochondria of cardiomyocytes were observed. Cardiomyocytes from R6/2 mice also showed abnormalities in mitochondrial structure (loss of longitudinal shape and changes in mitochondrial density) that may lead to deteriorations in energy metabolism [97]. The studies performed on this model have also shown that the hearts of these mice are characterized by decreased activity of the mammalian target of rapamycin kinase complex 1 (mTORC1) that could be the cause of the reduced heart weight which is observed in this strain as well as the lack of resistance to severe and chronic stress [98]. Nevertheless, Kojer et al. highlighted no changes in mitochondrial oxidative chain complexes in R6/2 mice hearts [78]. Mechanistically, pathological huntingtin was found to be responsible for cellular death via inhibition of the proteasome in the cytosol and the nucleus of cardiomyocytes of R6/2 mice [97]. On the other hand, some studies indicated the mHTT absence may be the underlying mechanism, as observed in HD mouse hearts [78,90]. Taken together, cardiac dysfunction in HD might derive not only from autonomic nervous system dysfunction but also from several cellular and tissue defects, such as energy metabolism impairment, associated with dysfunctional cardiac purine metabolism and signaling [99]. Figure 2 and Figure 3 illustrate changes in purine nucleotides metabolism and signaling in Huntington’s disease-related myopathies.

Research with HD patients detected reduced phosphocreatine to inorganic phosphate ratio in skeletal muscle of the symptomatic HD patients at rest (analyzed with a non-invasive 31P-MRS method). Moreover, muscle ATP/phosphocreatine and inorganic phosphate levels were significantly reduced in both symptomatic and presymptomatic HD subjects [100]. Furthermore, the maximum rate of mitochondrial ATP production during recovery from exercise was reduced by 44% in symptomatic HD patients and by 35% in presymptomatic HD carriers [77]. In the case of experimental models of HD, we have established that R6/2, as well as *Hdh*Q150 HD mice, exhibited decreased ATP, ADP, and AMP concentrations in *Extensor digitorum longus*, *Tibialis anterior*, and *Soleus*. Moreover, a significant reduction of phosphocreatine (PCr) and creatine (Cr) levels and the PCr/Cr ratio was observed in the examined skeletal muscles [75]. Similar changes were observed in HD mouse model hearts. We observed decreased concentrations of ATP and phosphocreatine as well as diminished ATP/ADP ratios. Interestingly, in contrast to skeletal muscles, this was accompanied by increased AMP levels [101]. Elevated concentration of cardiac AMP may activate AMP-regulated protein kinase (AMPK), which was shown as an increased AMPK phosphorylation status and enhanced AMPK mRNA transcript (observed also in HD-affected skeletal muscle) [102].

The functionality of the ATP-phosphocreatine shuttle, the transcriptional signature of genes involved in purine metabolism in HD-affected skeletal muscle and hearts were also assessed [101,102]. In the case of genes involved in purine nucleotide catabolism, skeletal muscle mRNA levels of Adenylate kinase 1 exhibited significant down-regulation while the mRNA levels of ecto-5′-nucleotidase remained unchanged in skeletal muscle (*Extensor digitorum longus*, *Tibialis anterior*, and *Soleus*). The same parameters were also studied in R6/2 mice hearts and demonstrated similar trends, such as reduced *Ak1* (Adenylate kinase 1) and no changes in *Nt5e* transcript levels [101]. Nevertheless, cardiac e5NT activity was significantly reduced [103]. Similarly, cardiac *Ampd3* (Adenosine monophosphate deaminase 3) transcript levels remained unchanged but AMPD activity was significantly reduced in R6/2 mice hearts, which could be caused by increased functional protein turnover or activity modulation in HD [101,103]. On the other hand, HD mouse skeletal muscle exhibited a significant up-regulation of the *Ampd3*, which could be caused by its alteration by denervation which is exclusively linked to skeletal muscle atrophy [104]. Moreover, HD mouse model skeletal muscles and hearts exhibited significant expression down-regulation of two genes involved in the purine nucleotide cycle-adenylosuccinate lyase (*Adsl*,) and adenylosuccinate lyase 1. Expression profiles of selected genes involved in the intracellular adenosine metabolism such as *Ada* (Adenosine deaminase) and *Adk* (Adenosine kinase) were also measured. Significant up-regulation for the cardiac *Ada* transcript (no changes in skeletal muscle *Ada* transcript) and in the cardiac and skeletal muscle transcription levels of adenosine kinase were noted. This agrees with observed increased ADA activity, inosine concertation, and reduced adenosine levels in HD mouse model hearts [103]. Furthermore, mRNA levels of *Pnp* (Purine nucleoside phosphorylase) as well as PNP activity, which degrades inosine, remain unchanged in HD mouse hearts, while *Pnp* transcript levels were significantly and uniformly up-regulated in each examined HD mouse skeletal muscle. This observation suggests intensive inosine degradation to hypoxanthine and its release from the skeletal muscle cell. Extracellular hypoxanthine influx might activate its degradation by other cells to xanthine and then uric acid by xanthine oxidase or xanthine dehydrogenase. Moreover, inhibition of xanthine oxidase should protect against muscle mass loss, thus its activation may implicate opposite effects [105]. Interestingly, the skeletal muscle transcript of *Xdh*-(Xanthine dehydrogenase) was significantly up-regulated while in heart was unaltered, which confirms our earlier thesis. Furthermore, up-regulation of the mRNA levels of one of the enzymes involved in purine synthesis de novo, such as phosphoribosylglycinamide formyltransferase (*Gart*), was noted in HD skeletal muscle. A different observation was obtained in *Hdh*Q150 mouse hearts, in whose *Gart* transcription was reduced. Nevertheless, cardiac and skeletal muscle transcript levels of other genes such as adenine phosphoribosyltransferase (*Aprt*) and amidophosphoribosyltransferase (*Ppat)* mRNA remained unchanged. Acceleration of purine de novo synthesis in HD skeletal muscle may indicate a greater demand for purine pool reconstruction in the cell. Probably, in HD mouse hearts, the signaling pathways aiming stabilized normal purine synthesis might be interrupted. Transcript rate analysis of genes involved in extracellular metabolism of purine nucleotides revealed significant down-regulation of *Entpd2* (Ectonucleoside triphosphate diphosphohydrolase 2) in investigated HD-affected skeletal muscle and heart. Nevertheless, eNTPD activity analysis did not reveal any changes in its cardiac activity in R6/2 mice.

In the case of purinergic signaling, there are still no literature data available regarding its role in skeletal muscle and heart dysfunction related to HD. Nevertheless, it is well known that enhanced expression of specific purinergic system elements for example in dystrophic muscles are important for dystrophic pathophysiology and could increase its severity considerably [106]. Dystrophin mutations in Duchene muscular dystrophy (DMD) coincide with significant P2X7 upregulation in muscle and alter receptor signaling in mouse dystrophic myoblasts and myofibers. Thus, P2X7 overexpression combined with the extracellular ATP-rich environment leads to cell dysfunction and death and ultimately to ineffective skeletal muscle regeneration [107]. Moreover, Ryten et al. demonstrated sequential expression of other purinergic receptors, such as P2X5, P2Y1, and P2X2 subtypes during the process of muscle regeneration [108]. Increased expression of the P2X1 receptor was reported in the aria of patients with dilated cardiomyopathy [109]. The protective role of P2X receptor activation (reduction of hypertrophy and increasing a life span) was highlighted in the calsequestrin overexpression model of cardiomyopathy [110,111]. Furthermore, roles for P2X7 receptors in dilated cardiomyopathy have been reported similarly to skeletal muscle pathologies [112]. Based on this data, the role of purinergic signaling in HD affected skeletal muscle and heart might be a new and interesting topic to delve into.

## 3. Purine Nucleotides Metabolism and Signaling as a Target for Huntington’s Disease Therapies

As presented in this review, there are many changes related to the purine metabolism in the central nervous system, skeletal muscle, and heart of HD patients, animals as well as cellular models. Nevertheless, these changes in those systems might involve different mechanisms. As described earlier, some studies indicated mHTT absence in HD mouse model hearts. Thus, the cardiac purine metabolism derangement could be caused only by HTT protein signaling dysfunction not by mutant protein accumulation, while in CNS and skeletal muscle purine metabolism impairment may be caused by loss of HTT protein function together with cellular mHTT accumulation. Deterioration in intracellular purine metabolism leads to the accumulation of purine metabolites that might be released from the cell via transporters. Indeed, we observed a significant accumulation of purine catabolites in HD mouse models and patients’ serum, which was strongly correlated with HD progression [101]. Nucleosides, as well as nucleotides released from the cell, might be degraded in the extracellular purine metabolism and activate purinergic signaling pathways. In summary, purine metabolism and signaling impairment might affect the development and progression of energy metabolism, signaling, and hence function derangements in HD-affected systems. Therefore, therapies leading to their improvement may have promising clinical implications in HD treatment.

Based on collected findings, therapeutic strategies may include compounds that directly correct disrupted ATP levels and lead to adenine nucleotide pool reconstitution in HD. Compounds, such as the coenzyme Q_10_ or creatinine, were widely tested and even investigated in clinical trials, but results were not promising [113]. An alternative might be the application of PPAR agonists, which have already undergone preclinical studies for the treatment of CNS, cardiovascular as well as skeletal muscle diseases. In 2016, the PPAR delta receptor agonist KD3010 was tested in the HD N171-82Q mouse model. Studies revealed improved motor function, reducing the progression of the neurodegenerative process and longer survival of treated animals [114]. Interestingly, studies performed by our team showed that simultaneous administration of adenosine metabolism inhibitors and substrates for adenine nucleotide synthesis improved mechanical functions of the heart, energy metabolism, and normalized adenine nucleotide pools [115,116], pointing at possible applications in HD treatment.

AMPD makes part of another altered purine pathway in HD. Skeletal muscles of the HD mouse model showed enhanced AMPD activity, while its activity in the heart was reduced. Pharmacological inhibition, as well as AMPD expression deletion in mice, led to a substantial enhancement of skeletal muscle contraction, induced mainly by AMP accumulation [117]. Moreover, AMPD inhibition seems to be also protective in cardiovascular diseases [118,119]. AMPD activity reduction noted in HD mouse model heart led to increased cardiac AMP levels and AMPK activation, but due to increased ADA activity, it was not capable of preventing intracellular adenosine pool depletion. Thus, in HD affected hearts adenosine levels augment, with suggested cardioprotective properties [120]. Interestingly, adenosine has been used to treat also epileptic diseases and seizures that are commonly observed in the juvenile form of HD. Thus, adenosine might be also a target for HD-affected CNS [121,122].

Furthermore, drugs increasing not only the intracellular but also the extracellular adenosine levels in HD-affected brain and heart might be protective. As described earlier, the most important enzyme that controls extracellular adenosine metabolism balance is eADA. It is well known that extracellular adenosine pool depletion is an important factor in the development of cardiovascular pathologies. Recently, we highlighted the therapeutic perspectives of eADA inhibition in the treatment of cardiovascular diseases such as atherosclerosis, myocardial ischemia-reperfusion injury, or hypertension [123]. Moreover, a study underlined that intrastriatal administration of ecto-nucleoside transporter (ENT) inhibitors increased the extracellular level of adenosine in the striatum of R6/2 mice to a much higher level, compared to controls, and improved HD mouse survival [124].

Impaired purinergic signaling in HD in CNS concerns mainly P2X7 and P2Y2 receptors. P2X7 antagonism in HD prevents neuronal death [67]. Nevertheless, only the P2X7 receptor seems to be important in skeletal muscle as well as heart failure. It is known that antagonizing or knocking out P2X7 or its downstream effectors, caspase-1 or NLRP3, in animal models decreased infarct size, improved cardiac function, and enhanced survival post myocardial infraction via reduced interelukin1β and intereukin18 levels in the heart [125,126,127,128]. Furthermore, genetic ablation and pharmacological inhibition of the P2X7 axis alleviated dystrophic phenotypes in mouse models of dystrophinopathy and sarcoglycanopathy [107]. Therefore, its antagonism may be a more suitable approach to treat not only HD-affected CNS system but also skeletal muscle and heart.

## Figures and Tables

**Figure 1 ijms-22-06545-f001:**
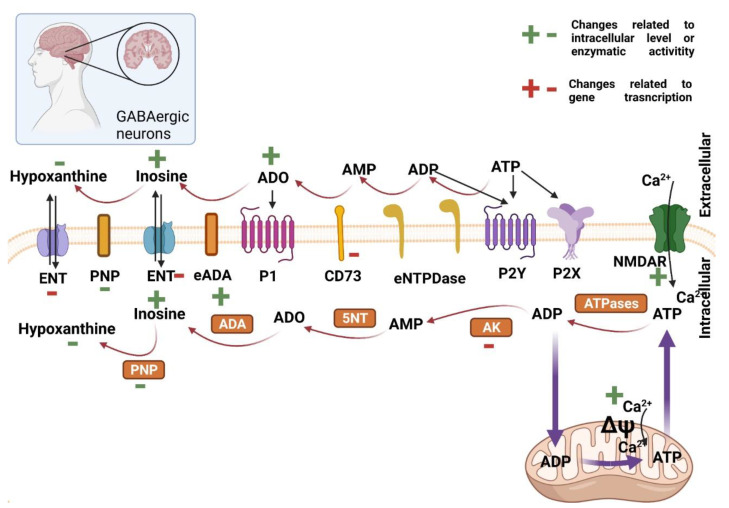
Purine metabolism and signaling in healthy and Huntington’s disease striatal GABAergic neurons. In the neurons of the striatum, ATP is heavily produced by mitochondria which also play important role in buffering cytosolic calcium signaling. In the cytosol, many enzymes (orange boxes) can convert ATP to AMP, ADO, IMP, inosine, and hypoxanthine. The latter ones can be transported to the extracellular milieu by ENT. Extracellularly, ATP can also be converted to inosine and hypoxanthine through a similar cascade by extracellular-faced enzymes anchored to the cell membrane. CD73 is the only extracellular 5NT. Pathophysiology of Huntington’s disease exerts alterations in the activity of enzymes (marked in green +/−) or by genomic expression (marked in red +/−). In Huntington’s disease condition, NMDA-glutamate receptors permit uncontrolled Ca^2+^ influx into cells. The excessive Ca^2+^ is transported to mitochondria, where it disrupts the membrane potential, releasing cytochrome c and promoting cell death. **P1**—Adenosine receptor; **P2X**—ATP receptor channel; **P2Y**—ATP/UTP receptor; **ATPases**—adenosine-5′-triphosphatases; **AK**—adenylate kinase; **eNTPD**—ecto-nucleoside triphosphate diphosphohydrolase; **5NT**—5′ nucleotidase; **CD73**—ecto-5′ nucleotidase; **eADA**—ecto-adenosine deaminase; **ADA**—adenosine deaminase; **PNP**—purine nucleoside phosphorylase; **ENT**—ecto nucleoside transporter. Created with BioRender.com (accessed on 5 May 20021).

**Figure 2 ijms-22-06545-f002:**
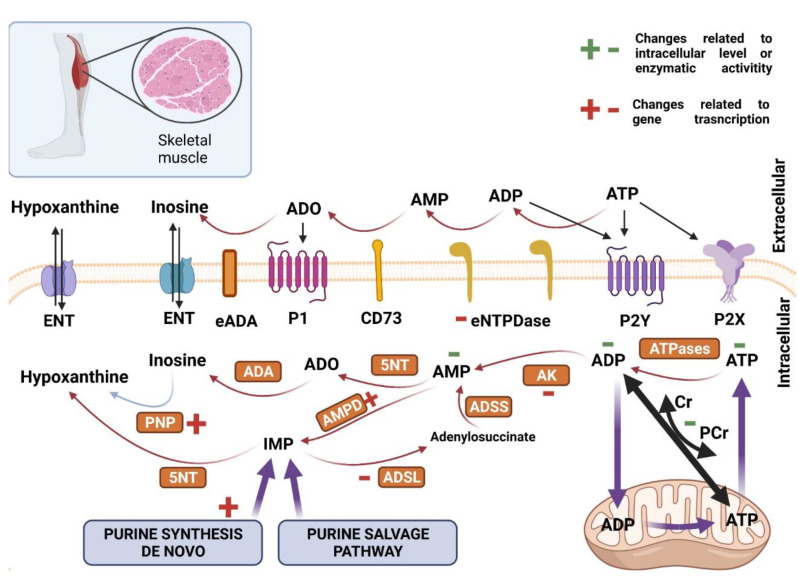
Purine metabolism and signaling in healthy and Huntington’s disease skeletal muscles. In the skeletal muscles, ATP is heavily produced by mitochondria and transported to the cytosol through the phosphocreatine shuttle. In the cytosol, many enzymes (orange boxes) can convert ATP to AMP, ADO, IMP, inosine, and hypoxanthine. The later ones can be transported to the extracellular milieu by ENT. Extracellularly, ATP can be converted to inosine through a similar cascade by extracellular-faced enzymes anchored to the cell membrane. CD73 is the only extracellular 5NT. Pathophysiology of Huntington’s disease exerts alterations in the activity of enzymes (marked in green +/−) or by genomic expression (marked in red +/−). **P1**—Adenosine receptor; **P2X**—ATP receptor channel; **P2Y**—ATP/UTP receptor; **ATPases**—adenosine-5′-triphosphatases; **AK**—adenylate kinase; **eNTPD**—ecto-nucleoside triphosphate diphosphohydrolase; **AMPD**—AMP deaminase; **5NT**—5′ nucleotidase; **CD73**—ecto-5′ nucleotidase; **eADA**—ecto-adenosine deaminase; **ADA**—adenosine deaminase; **PNP**—purine nucleoside phosphorylase; **ADSL**—adenylosuccinate lyase; **ADSS**—adenylosuccinate synthetase; **ENT**—ecto nucleoside transporter. Created with BioRender.com (accessed on 5 May 20021).

**Figure 3 ijms-22-06545-f003:**
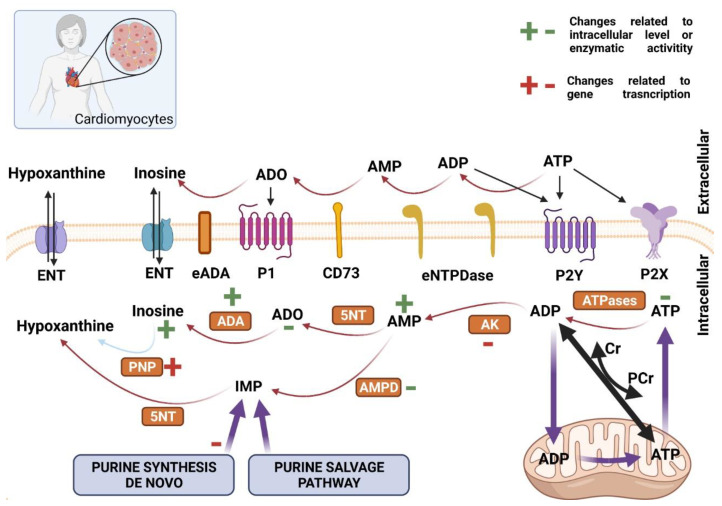
Purine metabolism and signaling in healthy and Huntington’s disease cardiomyocytes. In cardiomyocytes, ATP is heavily produced by mitochondria and transported to the cytosol through the phosphocreatine shuttle. In the cytosol, many enzymes (orange boxes) can convert ATP to AMP, ADO, IMP, inosine, and hypoxanthine. The later ones can be transported to the extracellular milieu by ENT. Extracellularly, ATP can also be converted to inosine through a similar cascade by extracellular-faced enzymes anchored to the cell membrane. CD73 is the only extracellular 5NT. Pathophysiology of Huntington’s disease exerts alterations in the activity of enzymes (marked in green +/−) or by genomic expression (marked in red +/−). **P1**—Adenosine receptor; **P2X**—ATP receptor channel; **P2Y**—ATP/UTP receptor; **ATPases**—adenosine-5′-triphosphatases; **AK**—adenylate kinase; **eNTPD**—ecto-nucleoside triphosphate diphosphohydrolase; **AMPD**—AMP deaminase; **5NT**—5′ nucleotidase; **CD73**—ecto-5′ nucleotidase; **eADA**—ecto-adenosine deaminase; **ADA**—adenosine deaminase; **PNP**—purine nucleoside phosphorylase; **ADSL**—adenylosuccinate lyase; **ADSS**—adenylosuccinate synthetase; **ENT**—ecto nucleoside transporter. Created with BioRender.com (accessed on 5 May 20021).

## Data Availability

No new data were created or analyzed in this study. Data sharing does not apply to this article.

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
