# Peer review of "Purine Nucleotides Metabolism and Signaling in Huntington’s Disease: Search for a Target for Novel Therapies"

_ijms, 2021, doi:10.3390/ijms22126545_

Round 1

Reviewer 1 Report

In the manuscripts, authors summarized the regulation of purine nucleotides metabolism and signaling in Huntington's disease. Although, the review is generally well written and covers main topics, there are small issues, the authors need to consider.

In the abstract and introduction, huntingtin gene should be written as italic (HTT). Moreover,  I would like to suggest that the author should need to add the information that CAG repeats translated to polyqlutamine stretch (the first sentence in abstract) to make clear for readers.

In the introduction part, the authors should briefly mention the onset of disease is correlated with the expansion of CAG repeats. (PMID:21163446).

In the part 2.1, the authors have suggested that HTT is not a fundamental protein in neuronal development. However, the previous study in human pluripotent stem cells suggested that loss of HTT can reduce the induction of neural and neuronal genes during differentiation (PMID: 30452683). The authors should provide this information and change this sentence.

Author Response

1.In the manuscripts, authors summarized the regulation of purine nucleotides metabolism and signaling in Huntington's disease. Although, the review is generally well written and covers main topics, there are small issues, the authors need to consider.

We thank the referee for their very positive comments in support of our work. 

2. In the abstract and introduction, huntingtin gene should be written as italic (HTT). 

We have changed the font accordingly (page 1, paragraph 11 and page 2, paragraph 50).

3. Moreover,  I would like to suggest that the author should need to add the information that CAG repeats translated to polyqlutamine stretch (the first sentence in abstract) to make clear for readers.

We thank reviewer for this suggestion and have placed this information in the abstract. (page 1, paragraphs 11-12).

4. In the introduction part, the authors should briefly mention the onset of disease is correlated with the expansion of CAG repeats. (PMID:21163446).

We have added this information with suggested reference (page 2, paragraph 55).

5. In the part 2.1, the authors have suggested that HTT is not a fundamental protein in neuronal development. However, the previous study in human pluripotent stem cells suggested that loss of HTT can reduce the induction of neural and neuronal genes during differentiation (PMID: 30452683). The authors should provide this information and change this sentence.

We have provided this information and changed this sentence (page 4, paragraphs 161-164).

Reviewer 2 Report

This is a very thorough and well-written review on the effects of Huntington's  disease on purine metabolism and activity. The authors provide a complete discussion of how these mechanisms are altered in Huntington's disease and may contribute to the pathophysiology of the disease. The paper also summarizes current efforts in the development and testing of potential therapies that may alter purine metabolism and treat the symptoms that relate to muscle damage from HD. The figures summarize the mechanisms discussed in the text and how they are altered in HD. This review will be worthwhile reading for the HD research community and I recommend publication. I do not have any changes to recommend prior to publication. 

Author Response

We thank the referee for their very positive comments in support of our work.